# CaSPR: Learning Canonical Spatiotemporal Point Cloud Representations

**Davis Rempe[1]    Tolga Birdal[1]    Yongheng Zhao[2]    Zan Gojcic[3]**
**Srinath Sridhar[1]    Leonidas J. Guibas[1]**
[1]Stanford University    [2]University of Padova    [3]ETH Zürich
geometry.stanford.edu/projects/caspr

## Abstract

We propose CaSPR, a method to learn object-centric **Ca**nonical **S**patiotemporal **P**oint Cloud **R**epresentations of dynamically moving or evolving objects. Our goal is to enable information aggregation over time and the interrogation of object state at any spatiotemporal neighborhood in the past, observed or not. Different from previous work, CaSPR learns representations that support spacetime continuity, are robust to variable and irregularly spacetime-sampled point clouds, and generalize to unseen object instances. Our approach divides the problem into two subtasks. First, we explicitly encode time by mapping an input point cloud sequence to a spatiotemporally-canonicalized object space. We then leverage this canonicalization to learn a spatiotemporal latent representation using neural ordinary differential equations and a generative model of dynamically evolving shapes using continuous normalizing flows. We demonstrate the effectiveness of our method on several applications including shape reconstruction, camera pose estimation, continuous spatiotemporal sequence reconstruction, and correspondence estimation from irregularly or intermittently sampled observations.

## 1   Introduction

The visible geometric properties of objects around us are constantly evolving over time due to object motion, articulation, deformation, or observer movement. Examples include the rigid *motion* of cars on the road, the *deformation* of clothes in the wind, and the *articulation* of moving humans. The ability to capture and reconstruct these spatiotemporally changing geometric object properties is critical in applications like autonomous driving, robotics, and mixed reality. Recent work has made progress on learning object shape representations from static 3D observations [49, 52, 53, 62, 73] and dynamic point clouds [9, 11, 40, 41, 45, 50, 80]. Yet, important limitations remain in terms of the lack of temporal continuity, robustness, and category-level generalization.

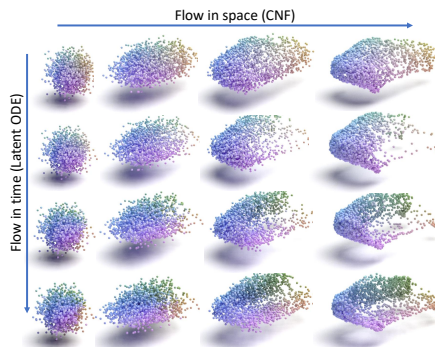

Figure 1: CaSPR builds a point cloud representation of (partially observed) objects continuously in both space ($x$-axis) and time ($y$-axis), while canonicalizing for extrinsic object properties like pose.

In this paper, we address the problem of learning object-centric representations that can aggregate and encode **spatiotemporal (ST) changes** in object shape as seen from a 3D sensor. This is challenging since dynamic point clouds captured by depth sensors or LIDAR are often incomplete and sparsely sampled over space and time. Furthermore, even point clouds corresponding to adjacent frames in a sequence will experience large sampling variation. Ideally, we would like spatiotemporal representations to satisfy several desirable properties. First, representations should allow us to

capture object shape **continuously** over space *and* time. They should encode changes in shape due to varying camera pose or temporal dynamics, and support shape generation at arbitrary spatiotemporal resolutions. Second, representations should be **robust** to irregular sampling patterns in space and time, including support for full or partial point clouds. Finally, representations should support within-category **generalization** to unseen object instances and to unseen temporal dynamics. While many of these properties are individually considered in prior work [11, 30, 41, 45, 68], a unified and rigorous treatment of all these factors in space and time is largely missing.

We address the limitations of previous work by learning a novel object-centric ST representation which satisfies the above properties. To this end, we introduce **CaSPR** – a method to learn **Ca**nonical **S**patiotemporal **P**oint Cloud **R**epresentations. In our approach, we split the task into two: (1) *canonicalizing* an input object point cloud sequence (partial or complete) into a shared 4D container space, and (2) learning a continuous ST latent representation on top of this canonicalized space. For the former, we build upon the Normalized Object Coordinate Space (NOCS) [63, 72] which canonicalizes intra-class 3D shape variation by normalizing for extrinsic properties like position, orientation, and scale. We extend NOCS to a 4D **Temporal-NOCS (T-NOCS)**, which additionally normalizes the duration of the input sequence to a unit interval. Given dynamic point cloud sequences, our ST canonicalization yields spacetime-normalized point clouds. In Sec. 5, we show that this allows learning representations that generalize to novel shapes and dynamics.

We learn ST representations of canonicalized point clouds using *Neural Ordinary Differential Equations* (Neural ODEs) [9]. Different from previous work, we use a **Latent ODE** that operates in a lower-dimensional *learned latent space* which increases efficiency while still capturing object shape dynamics. Given an input sequence, the canonicalization network and Latent ODE together extract features that constitute an ST representation. To continuously *generate* novel spatiotemporal point clouds conditioned on an input sequence, we further leverage invertible *Continuous Normalizing Flows* (CNFs) [6, 24] which transform Gaussian noise directly to the visible part of an object's shape at a desired timestep. Besides continuity, CNFs provide direct likelihood evaluation which we use as a training loss. Together, as shown in Fig. 1, the Latent ODE and CNF constitute a generative model that is continuous in spacetime and robust to sparse and varied inputs. Unlike previous work [11, 41], our approach is continuous and explicitly avoids treating time as another spatial dimension by respecting its unique aspects (*e.g.*, unidirectionality).

We demonstrate that CaSPR is useful in numerous applications including (1) continuous spacetime shape reconstruction from sparse, partial, or temporally non-uniform input point cloud sequences, (2) spatiotemporal 6D pose estimation, and (3) information propagation via space-time correspondences under rigid or non-rigid transformations. Our experiments show improvements to previous work while also providing insights on the emergence of intra-class shape correspondence and the learning of *time unidirectionality* [19]. In summary, our contributions are:

1. The CaSPR encoder network that consumes dynamic object point cloud sequences and canonicalizes them to normalized spacetime (T-NOCS).
2. The CaSPR representation of canonicalized point clouds using a Latent ODE to explicitly encode temporal dynamics, and an associated CNF for generating shapes continuously in spacetime.
3. A diverse set of applications of this technique, including partial or full shape reconstruction, spatiotemporal sequence recovery, camera pose estimation, and correspondence estimation.

## 2   Related Work

**Neural Representations of Point Sets**   Advances in 2D deep architectures leapt into the realm of point clouds with PointNet [52]. The lack of locality in PointNet was later addressed by a diverse set of works [16, 17, 38, 60, 64, 67, 74, 77, 83], including PointNet++ [53] – a permutation invariant architecture capable of learning both local and global point features. We refer the reader to Guo *et al.* [28] for a thorough review. Treating time as the fourth dimension, our method heavily leverages propositions from these works. Continuous reconstruction of an object's spatial geometry has been explored by recent works in learning implicit shape representations [10, 29, 43, 49].

**Spatiotemporal Networks for 3D Data**   Analogous to volumetric 3D convolutions on video frames [36, 69, 82], a direct way to process spatiotemporal point cloud data is performing 4D convolutions on a voxel representation. This poses three challenges: (1) storing 4D volumes densely is inefficient and impractical, (2) direct correlation of spatial and temporal distances is undesirable,

and (3) the inability to account for timestamps can hinder the final performance. These challenges have fostered further research along multiple fronts. For example, a large body of works [3, 27, 40, 75] has addressed temporal changes between a pair of scans as per-point displacements or *scene flow* [70]. While representing dynamics as fields of change is tempting, such methods lack an explicit notion of time. MeteorNet [41] was an early work to learn flow on raw point cloud sequences, however it requires explicit local ST neighborhoods which is undesirable for accuracy and generalization. Prant *et al.* [50] use temporal frames as a cue of coherence to stabilize the generation of points. CloudLSTM [80] models temporal dependencies implicitly within sequence-to-sequence learning. Making use of time in a more direct fashion, MinkowskiNet [11] proposed an efficient ST 4D CNN to exploit the sparsity of point sets. This method can efficiently perform 4D sparse convolutions, but can neither canonicalize time nor perform ST aggregation. OccupancyFlow [45] used occupancy networks [43] and Neural ODEs [9] to have an explicit notion of time.

Our method can be viewed as learning the underlying *kinematic spacetime surface* of an object motion: an idea from traditional computer vision literature for dynamic geometry registration [44].

**Canonicalization**    Regressing 3D points in a common global reference frame dates back to 6D camera relocalization and is known as *scene coordinates* [61]. In the context of learning the *normalized object coordinate space* (NOCS), [72] is notable for explicitly mapping the input to canonical *object coordinates*. Thanks to this normalization, NOCS enabled category-level pose estimation and has been extended to articulated objects [37], category-level rigid 3D reconstruction [12, 25, 31] via multiview aggregation [63], and non-rigid shape reconstruction either via deep implicit surfaces [79] or by disentangling viewpoint and deformation [46]. Chen *et al.* [7] proposed a latent variational NOCS to *generate* points in a canonical frame.

**Normalizing Flows and Neural ODEs**    The idea of transforming noise into data dates back to whitening transforms [22] and Gaussianization [8]. Tabak and Turner [66] officially defined normalizing flows (NFs) as the composition of simple maps and used it for non-parametric density estimation. NFs were immediately extended to deep networks and high dimensional data by Rippel and Adams [56]. Rezende and Mohamed used NFs in the setting of variational inference [54] and popularized them as a standalone tool for deep generative modeling *e.g.* [32, 65]. Thanks to their invertibility and exact likelihood estimation, NFs are now prevalent and have been explored in the context of graph neural networks [39], generative adversarial networks [26], bypassing topological limitations [2, 14, 18], flows on Riemannian manifolds [23, 42, 59], equivariant flows [4, 34, 55], and connections to optimal transport [20, 47, 71, 81]. The limit case where the sequence of transformations are indexed by real numbers yields continuous-time flows: the celebrated Neural ODEs [6], their latent counterparts [57], and FFJORD [24], an invertible generative model with unbiased density estimation. For a comprehensive review, we refer the reader to the concurrent surveys of [33, 48].

Our algorithm is highly connected to PointFlow [76] and C-Flow [51]. However, we tackle encoding and generating spatiotemporal point sets in addition to canonicalization while both of these works use CNFs in generative modeling of 3D point sets without canonicalizing.

## 3    Background

In this section, we lay out the notation and mathematical background required in Sec. 4.

**Definition 1 (Flow & Trajectory)** *Let us define a $d$-dimensional **flow** to be a parametric family of homeomorphisms $\phi : \mathcal{M} \times \mathbb{R} \mapsto \mathcal{M}$ acting on a vector $\mathbf{z} \in \mathcal{M} \subset \mathbb{R}^d$ with $\phi_0(\mathbf{z}) = \mathbf{z}$ (identity map) and $\phi_t(\mathbf{z}) = \mathbf{z}_t$. A temporal subspace of flows is said to be a **trajectory** $\mathcal{T}(\mathbf{z}) = \{\phi_t(\mathbf{z})\}_t$ if $\mathcal{T}(\mathbf{z}) \cap \mathcal{T}(\mathbf{y}) = \emptyset$ for all $\mathbf{z} \neq \mathbf{y}$, i.e., different trajectories never intersect [13, 18].*

**Definition 2 (ODE-Flow, Neural ODE & Latent ODE)** *For any given flow $\phi$ there exists a corresponding ordinary differential equation (**ODE**) constructed by attaching an optionally time-dependent vector $f(\mathbf{z}, t) \in \mathbb{R}^d$ to every point $\mathbf{z} \in \mathcal{M}$ resulting in a vector field s.t. $f(\mathbf{z}) = \phi'(\mathbf{z})|_{t=0}$. Starting from the initial state $\mathbf{z}_0$, this ODE given by $\frac{dz(t)}{dt} = f(z(t), t)$ can be integrated for time $T$ modeling the flow $\phi_{t=T}$:*

$$\mathbf{z}_T = \phi_T(\mathbf{z}_0) = \mathbf{z}_0 + \int_0^T f_\theta(\mathbf{z}_t, t) \, dt, \tag{1}$$

*where $\mathbf{z}_t \triangleq z(t)$ and the field $f$ is parameterized by $\boldsymbol{\theta} = \{\theta_i\}_i$. By the Picard–Lindelöf theorem [13], if $f$ is continuously differentiable then the initial value problem in Eq (1) has a unique solution.*

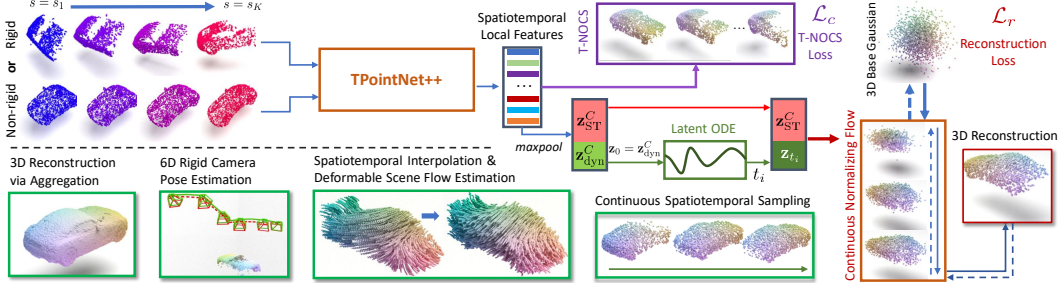

Figure 2: Architecture and applications of CaSPR. Our model consumes rigid or deformable point cloud sequences and maps them to a spatiotemporal canonical latent space whose coordinates are visualized by RGB colors (purple box). Using a Latent ODE, it advects a latent subspace forward in time to model temporal dynamics. A continuous normalizing flow [24] (shown in red) decodes the final latent code to 3D space by mapping Gaussian noise to the partial or full shape at desired timesteps. CaSPR enables multiple applications shown in green boxes. Training directions for the normalizing flow are indicated by dashed arrows.

*Instead of handcrafting, **Neural ODEs** [9] seek a function $f$ that suits a given objective by modeling $f$ as a neural network. We refer to a Neural ODE operating in a latent space as a **Latent ODE**.*

Numerous forms of Neural ODEs model $f(\cdot)$ to be *autonomous*, *i.e.*, time independent $f(\mathbf{z}_t) \equiv f(\mathbf{z}_t, t)$ [9, 18, 57], whose output fully characterizes the trajectory. While a Neural ODE advects single particles, **generative modeling** approximates the full target probability density which requires expressive models capable of exact density evaluation and sampling that avoids mode collapse.

**Definition 3 (Continuous Normalizing Flow (CNF))** *Starting from a simple $d_{\mathcal{B}}$-dimensional base distribution $p_y$ with $\mathbf{y}_0 \in \mathbb{R}^{d_{\mathcal{B}}} \sim p_y(\mathbf{y})$, **CNFs** [6, 24] aim to approximate the complex target distribution $p_x(\mathbf{x})$ by bijectively mapping empirical samples of the target to the base using an invertible function $g_\beta : \mathbb{R}^{d_{\mathcal{B}}} \mapsto \mathbb{R}^{d_{\mathcal{B}}}$ with parameters $\boldsymbol{\beta} = \{\beta_i\}_i$. Then the probability density function transforms with respect to the change of variables: $\log p_x(\mathbf{x}) = \log p_y(\mathbf{y}) - \log \det \nabla g_\beta(\mathbf{y})$. The warping function $g$ can be replaced by an integral of continuous-time dynamics yielding a form similar to Neural ODEs except that we now consider distributions [24]:*

$$\log p_x(\mathbf{x}) = \log p_y(\mathbf{y}_0) - \int_0^T \text{Tr}\left(\frac{\partial g_\beta(\mathbf{y}_t, t \mid \mathbf{z})}{\partial \mathbf{y}_t}\right) dt, \tag{2}$$

*with the simplest choice that the base distribution $\mathbf{y}_0$ is in a $d$-dimensional ball, $p_y \sim \mathcal{N}(\mathbf{0}, \mathbf{I})$. Here $\mathbf{z} \in \mathbb{R}^d$ is an optional **conditioning** latent vector [76]. Note that this continuous system is non-autonomous i.e., time varying and every non-autonomous system can be converted to an autonomous one by raising the dimension to include time [15, 18].*

## 4 Method

We consider as input a sequence of potentially partial, clutter-free 3D scans (readily captured by depth sensors or LIDAR) of an object belonging to a known category. This observation is represented as a point cloud $\mathcal{X} = \{\mathbf{x}_i \in \mathbb{R}^3 = \{x_i, y_i, z_i\} \mid i = 1, \dots, M'\}$. For a sequence of $K$ potentially non-uniformly sampled timesteps, we denote a **spatiotemporal** (ST) point cloud as $\mathcal{P} = \{\mathcal{P}_k\}_{k=1}^K$, where $\mathcal{P}_k = \{\mathbf{p}_i \in \mathbb{R}^4 = \{x_i, y_i, z_i, s_k\} \mid i = 1, \dots, M_k\}$, $M_k$ is the number of points at frame $k \in [1, K]$ and at the *time* $s_k \in [s_1, s_K] \subset \mathbb{R}$ with $M = \sum_{k=1}^K M_k$. Our goal is to explain $\mathcal{P}$ by learning a *continuous representation* of shape that is invariant to extrinsic properties while aggregating intrinsic properties along the direction of time. **CaSPR** achieves this through:

1. A **canonical** spacetime container where extrinsic properties such as object pose are factored out,
2. A continuous **latent** representation which can be queried at arbitrary spacetime steps, and
3. A **generative** model capable of reconstructing partial observations conditioned on a latent code.

We first describe the method design for each of these components, depicted in Fig. 2, followed by implementation and architectural details in Sec. 4.1.

**Canonicalization**: The first step is *canonicalization* of a 4D ST point cloud sequence with the goal of associating observations at different time steps to a common canonical space. Unlike prior work

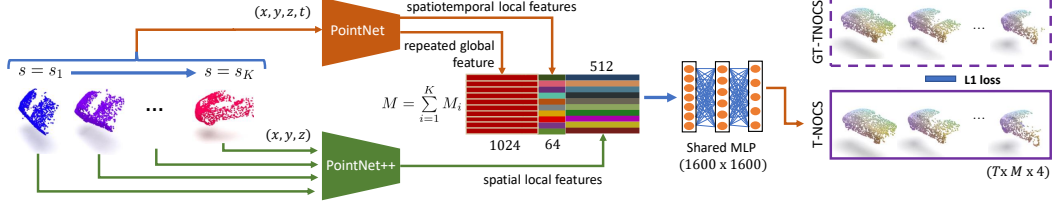

Figure 3: Architecture of our ST point-set canonicalization network, *TPointNet++*. It uses two branches that extract ST features using a 4D PointNet and per-view 3D local features via PointNet++. These features are combined and passed to an MLP to regress the T-NOCS points. Training is supervised via GT coordinates.

which assumes already-canonical inputs [45, 76], this step allows CaSPR to operate on raw point cloud sequences in world space and enables multiple applications (see Fig. 2). Other previous work has considered canonicalization of extrinsic properties from RGB images [63, 72] or a 3D point cloud [37], but our method operates on a 4D point cloud and explicitly accounts for time labels. Our goal is to find an injective *spacetime canonicalizer* $c_\alpha(\cdot) : \mathcal{P} \mapsto \overline{\mathcal{P}} \times \overline{\mathcal{Z}}$ parameterized by $\boldsymbol{\alpha} = \{\alpha_i\}_i$, that maps a point cloud sequence $\mathcal{P}$ to a canonical *unit tesseract* $\overline{\mathcal{P}} = \{\overline{\mathcal{P}}_k\}_{k=1}^K$, where $\overline{\mathcal{P}}_k = \{\overline{\mathbf{p}}_i \in \mathbb{R}^4 = \{\bar{x}_i, \bar{y}_i, \bar{z}_i, \bar{t}_k\} \in [0,1] \,|\, i = 1, \ldots, M_k\}$ and $\mathbf{z}^C \in \overline{\mathcal{Z}} \subset \mathbb{R}^d$ is the corresponding canonical *latent* representation (embedding) of the sequence. Note that in addition to position and orientation, $\overline{\mathcal{P}}$ is normalized to have time in unit duration. We refer to $\overline{\mathcal{P}}$ as **Temporal-NOCS** (**T-NOCS**) as it extends NOCS [63, 72]. T-NOCS points are visualized using the spatial coordinate as the *RGB* color in Fig. 2 and 3. Given a 4D point cloud in the world frame, we can aggregate the entire shape from $K$ partial views by a simple union: $\overline{\mathcal{P}} = \bigcup_{i=1}^K c_\alpha(\mathcal{P}_i)$ [63]. Moreover, due to its injectivity, $c_\alpha(\cdot)$ preserves *correspondences*, a property useful in tasks like pose estimation or label propagation. We outline the details and challenges involved in designing a canonicalizer in Sec. 4.1.

**Continuous Spatiotemporal Representation**: While a global ST latent embedding is beneficial for canonicalization and aggregation of partial point clouds, we are interested in continuously modeling the ST input, *i.e.*, being able to compute a representation for unobserved timesteps at arbitrary spacetime resolutions. To achieve this, we split the latent representation: $\mathbf{z}^C = [\mathbf{z}_{\text{ST}}^C, \mathbf{z}_{\text{dyn}}^C]$ where $\mathbf{z}_{\text{ST}}^C$ is the *static* ST descriptor and $\mathbf{z}_{\text{dyn}}^C$ is used to initialize an autonomous Latent ODE $\frac{d\mathbf{z}_t}{dt} = f_\theta(\mathbf{z}_t)$ as described in Dfn. 2: $\mathbf{z}_0 = \mathbf{z}_{\text{dyn}}^C \in \mathbb{R}^d$. We choose to advect the ODE in the latent space (rather than physical space [45]) to (1) enable *learning* a space best-suited to modeling the dynamics of the observed data, and (2) improve scalability due to the fixed feature size. Due to the time-independence of $f_\theta$, $\mathbf{z}_0$ fully characterizes the latent trajectory. Advecting $\mathbf{z}_0$ forward in time by solving this ODE until any canonical timestamp $T \leq 1$ yields a continuous representation in time $\mathbf{z}_T$ that can explain changing object properties. We finally obtain a dynamic spatiotemporal representation in the product space: $\mathbf{z} \in \mathbb{R}^D = [\mathbf{z}_{\text{ST}}^C, \mathbf{z}_T]$. Due to canonicalization to the unit interval, $T > 1$ implies *extrapolation*.

**Spatiotemporal Generative Model**: Numerous methods exist for point set generation [1, 25, 83], but most are not suited for sampling on the surface of a partial 4D ST point cloud. Therefore, we adapt CNFs [24, 76] as defined in Sec. 3. To generate a novel ST shape, *i.e.*, a sequence of 3D shapes $\mathcal{X}_1 \ldots \mathcal{X}_K$, we simulate the Latent ODE for $t = 0 \ldots T$ and obtain representations for each of the canonical shapes in the sequence: $\mathbf{z}_{t=0} \cdots \mathbf{z}_{t=T}$. We then sample the base distribution $\mathbf{y}_k \in \mathbb{R}^{d_\mathcal{B}=3} \sim p_y(\mathbf{y}) \triangleq \mathcal{N}(\mathbf{0}, \mathbf{I})$ and evaluate the conditional CNF in Eq (2) by passing each sample $\mathbf{y}_k$ through the flow $g_\beta(\mathbf{y}_k \,|\, \mathbf{z}_t)$ conditioned on $\mathbf{z}_t$. Note that the flow is time dependent, *i.e.*, non-autonomous. To increase the temporal resolution of the output samples we pick the timesteps with higher frequency, whereas to *densify* spatially, we simply generate more samples $\mathbf{y}_k$.

## 4.1 Network Architecture

We now detail our implementations of the canonicalizer $c_\alpha$, Latent ODE network $f_\theta$, and CNF $g_\beta$.

**TPointNet++** $c_\alpha(\cdot)$: The design of our canonicalizer is influenced by (1) the desire to avoid ST neighborhood queries, (2) to treat time as important as the spatial dimensions, and (3) injecting how an object appears during motion in space into its local descriptors resulting in more expressive features. While it is tempting to directly apply existing point cloud architectures such as PointNet [52] or PointNet++ [53], we found experimentally that they were individually insufficient (*c.f.* Sec. 5). To meet our goals, we instead introduce a hybrid *TPointNet++* architecture as shown in Fig. 3 to implement $c_\alpha$ and canonicalize $\mathcal{P}$ to $\overline{\mathcal{P}}$. TPointNet++ contains a PointNet branch that consumes the

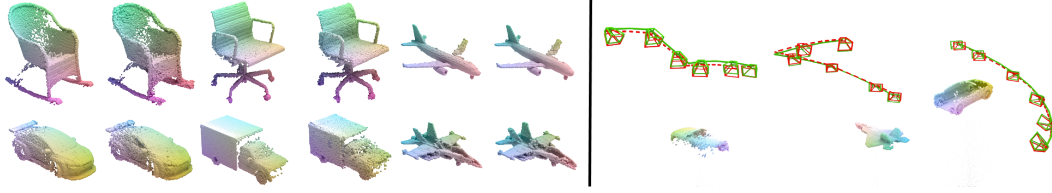

Figure 4: Canonicalization applications. Partial shape reconstruction (left section) shows pairs of GT (left) and predicted shapes (right). Pose estimation (right section) shows GT (green, solid) and predicted (red, dashed) camera pose based on regressed T-NOCS points. Points are colored by their T-NOCS location.

entire 4D point cloud to extract both a 1024-dimensional global feature and 64-dimensional per-point ST features. This treats time explicitly and equally to each spatial dimension. We also use PointNet++ to extract a 512-dimensional local feature at each input point by applying it at each cross-section in time with no timestamp. We feed all features into a shared multi-layer perceptron (MLP) to arrive at 1600-dimensional embeddings corresponding to each input point.

We use the pointwise embeddings in two ways: (1) they are passed through a shared linear layer followed by a *sigmoid* function to estimate the T-NOCS coordinates $\widehat{\mathcal{P}}$ which approximate the ground truth $\overline{\mathcal{P}}$, and (2) we max-pool all per-point features into a single latent representation of T-NOCS $\mathbf{z}^C \in \mathbb{R}^{1600}$ which is used by the Latent ODE and CNF as described below. The full canonicalizer $c_\alpha(\cdot)$ can be trained independently for T-NOCS regression, or jointly with a downstream task.

**Latent ODE $f_\theta(\cdot)$ and Reconstruction CNF $g_\beta(\cdot)$:** The full CaSPR architecture is depicted in Fig. 2. It builds upon the embedding from TPointNet++ by first splitting it into two parts $\mathbf{z}^C = [\mathbf{z}_{\text{ST}}^C, \mathbf{z}_0 \triangleq \mathbf{z}_{\text{dyn}}^C]$. The dynamics network of the Latent ODE $f_\theta$ is an MLP with three hidden layers of size 512. We use a Runge-Kutta 4(5) solver [35, 58] with adaptive step sizes which supports backpropagation using the adjoint method [9]. The static feature, $\mathbf{z}_{\text{ST}}^C \in \mathbb{R}^{1536}$ is skip-connected and concatenated with $\mathbf{z}_T$ to yield $\mathbf{z} \in \mathbb{R}^{1600}$ which conditions the reconstruction at $t = T$.

To sample the surface represented by $\mathbf{z}$, we use a FFJORD conditional-CNF [24, 76] as explained in Sec. 3 and 4 to map 3D Gaussian noise $\mathbf{y}_0 \in \mathbb{R}^{d_\mathcal{B}=3} \sim \mathcal{N}(\mathbf{0}, \mathbf{I})$ onto the shape surface. The dynamics of this flow $g_\beta(\mathbf{y}_t, t \mid \mathbf{z})$ are learned with a modified MLP [24] which leverages a gating mechanism at each layer to inject information about the current context including $\mathbf{z}$ and current time $t$ of the flow. This MLP contains three hidden layers of size 512, and we use the same solver as the Latent ODE. Please refer to the supplement for additional architectural details.

**Training and Inference:** CaSPR is trained with two objectives that use the GT canonical point cloud sequence $\overline{\mathcal{P}}$ as supervision. We primarily seek to maximize the *log-likelihood* of canonical spatial points on the surface of the object when mapped to the base Gaussian using the CNF. This reconstruction loss is $\mathcal{L}_r = -\sum_{k=1}^{K} \sum_{i=1}^{M_k} \log p_x(\bar{\mathbf{x}}_i \mid \mathbf{z}_{t_k})$ where $\bar{\mathbf{x}}_i$ is the spatial part of $\bar{\mathbf{p}}_i \in \overline{\mathcal{P}}_k$ and the log-likelihood is computed using Eq (2). Secondly, we supervise the T-NOCS predictions from TPointNet++ with an L1 loss $\mathcal{L}_c = \sum_{i=1}^{M} |\widehat{\mathbf{p}}_i - \bar{\mathbf{p}}_i|$ with $\bar{\mathbf{p}}_i \in \overline{\mathcal{P}}$ and $\widehat{\mathbf{p}}_i \in \widehat{\mathcal{P}}$. We jointly train TPointNet++, the Latent ODE, and CNF for $\alpha$, $\theta$ and $\beta$ respectively with the final loss $\mathcal{L} = \mathcal{L}_r + \mathcal{L}_c$. During inference, TPointNet++ processes a raw point cloud sequence of an unseen shape and motion to obtain the ST embedding and canonicalized T-NOCS points. The Latent ODE, initialized by this embedding, is solved forward in time to any number of canonical "query" timestamps. For each timestamp, the Latent ODE produces the feature to condition the CNF which reconstructs the object surface by the forward flow of Gaussian samples. The combined continuity of the Latent ODE and CNF enables CaSPR to reconstruct the input sequence at any desired ST resolution.

## 5 Experimental Evaluations

We now evaluate the canonicalization, representation, and reconstruction capabilities of CaSPR, demonstrate its utility in multiple downstream tasks, and justify design choices.

**Dataset and Preprocessing:** We introduce a new dataset containing simulated rigid motion of objects in three ShapeNet [5] categories: cars, chairs, and airplanes. The motion is produced with randomly generated camera trajectories (Fig. 4) and allows us to obtain the necessary inputs and supervision for CaSPR: sequences of raw partial point clouds from depth maps with corresponding canonical T-NOCS point clouds. Each sequence contains $K = 10$ frames with associated timestamps.

Table 2: Partial surface sequence reconstruction. Chamfer (CD) and Earth Mover's Distances (EMD) are multiplied by $10^3$. On the left (*10 Observed*), 10 frames are given as input and all are reconstructed. On the right, 3 frames are used as input (*3 Observed*), but methods also reconstruct intermediate unseen steps (*7 Unobserved*).

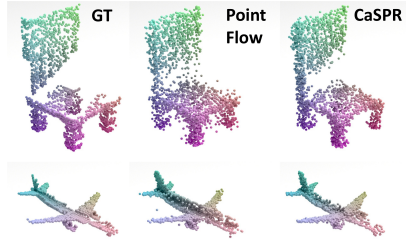

| Method | Category | 10 Observed CD | 10 Observed EMD | 3 Observed CD | 3 Observed EMD | 7 Unobserved CD | 7 Unobserved EMD |
|---|---|---|---|---|---|---|---|
| PointFlow | Cars | **0.454** | 12.838 | **0.455** | 12.743 | **0.525** | 13.911 |
| CaSPR-Atlas | | 0.492 | 19.528 | 0.540 | 22.099 | 0.530 | 19.635 |
| CaSPR | | 0.566 | **10.103** | 0.590 | **11.464** | 0.584 | **11.259** |
| PointFlow | Chairs | 0.799 | 17.267 | 0.796 | 17.294 | 0.950 | 18.442 |
| CaSPR-Atlas | | **0.706** | 48.665 | 0.723 | 48.912 | 0.749 | 47.322 |
| CaSPR | | 0.715 | **13.009** | **0.681** | **13.310** | **0.683** | **13.564** |
| PointFlow | Airplanes | 0.251 | 9.500 | 0.252 | 9.534 | 0.281 | 9.814 |
| CaSPR-Atlas | | 0.237 | 18.827 | 0.255 | 18.525 | 0.269 | 17.933 |
| CaSPR | | **0.231** | **6.026** | **0.215** | **6.144** | **0.216** | **6.175** |

Figure 5: Reconstruction results. CaSPR accurately captures occlusion boundaries for camera motion at observed and unobserved timesteps, unlike linear feature interpolation with PointFlow.

Raw point cloud sequences are labeled with uniform timestamps from $s_1 = 0.0$ to $s_K = 5.0$ while canonicalized timestamps range from $\bar{t}_1 = 0$ to $\bar{t}_K = 1$. For training, 5 frames with 1024 points are randomly subsampled from each sequence, giving non-uniform step sizes between observations. At test time, we use a *different spatiotemporal sampling* for sequences of held-out object instances: all 10 frames, each with 2048 points. Separate CaSPR models are trained for each shape category.

**Evaluation Procedure**: To measure canonicalization errors, T-NOCS coordinates are split into the spatial and temporal part with GT given by $\bar{\mathcal{X}}$ and $\bar{\mathbf{t}}$ respectively. The *spatial error* at frame $k$ is $\frac{1}{M_k} \sum_{i=1}^{M_k} \|\hat{\mathbf{x}}_i - \bar{\mathbf{x}}_i\|_2$ and the *temporal error* is $\frac{1}{M_k} \sum_{i=1}^{M_k} |\hat{t}_i - \bar{t}_i|$ . For reconstruction, the Chamfer Distance (CD) and Earth Mover's Distance (EMD) are measured (and reported multiplied by $10^3$). Lower is better for all metrics; we report the median over all test frames because outlier shapes cause less informative mean errors. Unless stated otherwise, qualitative point cloud results (*e.g.*, Fig. 4) are colored by their canonical coordinate values (so corresponding points should have the same color).

### 5.1 Evaluations and Applications

**Canonicalization**: We first evaluate the accuracy of canonicalizing raw partial point cloud sequences to T-NOCS using TPointNet++. Tab. 1 shows median errors over all frames in the test set. The bottom part evaluates TPointNet++ on each shape category while the top compares with baselines on cars (please see supplementary for more details). No-

Table 1: Canonicalization performance.

| Method | Category | Spatial Err | Time Err |
|---|---|---|---|
| MeteorNet | Cars | 0.0633 | **0.0001** |
| PointNet++ No Time | | 0.0530 | — |
| PointNet++ w/ Time | | 0.0510 | 0.0005 |
| PointNet | | 0.0250 | 0.0012 |
| TPointNet++ No Time | | 0.0122 | — |
| TPointNet++ | Cars | **0.0118** | 0.0011 |
| TPointNet++ | Chairs | 0.0102 | 0.0008 |
| TPointNet++ | Airplanes | 0.0064 | 0.0009 |

tably, for spatial prediction, TPointNet++ outperforms variations of both PointNet [52] and PointNet++ [53], along with their spatiotemporal extension MeteorNet [41]. This indicates that our ST design yields more distinctive features both spatially and temporally. MeteorNet and PointNet++ (with time) achieve impressive time errors thanks to skip connections that pass the input timestamps directly towards the end of the network. Qualitative results of canonicalization are in Fig. 4.

**Representation and Reconstruction**: We evaluate CaSPR's ability to represent and reconstruct observed and unobserved frames of *raw* partial point cloud sequences. The full model is trained on each category separately using both $\mathcal{L}_r$ and $\mathcal{L}_c$, and is compared to two baselines. The first is a variation of CaSPR where the CNF is replaced with an AtlasNet [25] decoder using 64 patches – an alternative approach to achieve spatial continuity. This model is trained with $\mathcal{L}_c$ and a CD loss (rather than $\mathcal{L}_r$). The second baseline is the deterministic PointFlow [76] autoencoder trained to reconstruct a *single canonical* partial point cloud. This model operates on a single timestep and receives the **already canonical** point cloud as input: an easier problem. We achieve temporal continuity with PointFlow by first encoding a pair of adjacent observed point clouds to derive two shape features, and then linearly interpolating to the desired timestamp – one alternative to attain temporal continuity. The interpolated feature conditions PointFlow's CNF to sample the partial surface, similar to CaSPR.

Tab. 2 reports median CD and EMD at reconstructed test steps for each method. We evaluate two cases: (1) models receive and reconstruct all 10 observed frames (left), and (2) models get the first, middle, and last steps of a sequence and reconstruct both these 3 observed and 7 unobserved frames (right). CaSPR outperforms PointFlow in most cases, even at observed timesteps, despite operating on raw point clouds in the world frame instead of canonical. Because PointFlow reconstructs each

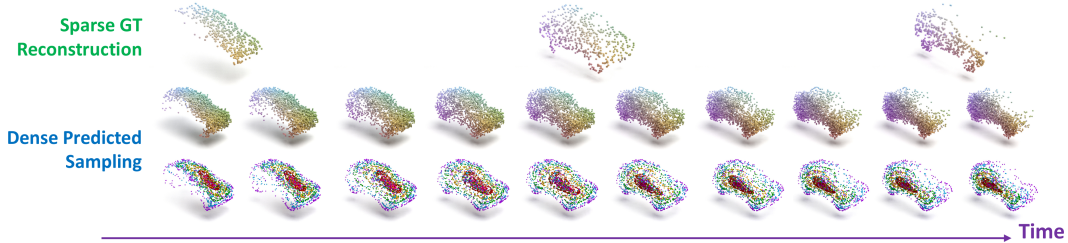

**Sparse GT Reconstruction**

**Dense Predicted Sampling**

**Time**

Figure 7: Continuous interpolation results. From three sparse frames of input with GT canonical points shown on top, CaSPR reconstructs the sequence more densely in space and time (middle). Contours of the Gaussian flowed to the car surface are shown on bottom (red is highest probability).

frame independently, it lacks temporal context resulting in degraded occlusion boundaries (Fig. 5) and thus higher EMD. CaSPR gives consistent errors across observed and unobserved frames due to the learned motion prior of the Latent ODE, in contrast to linear feature interpolation that sees a marked performance drop for unobserved frames. The AtlasNet decoder achieves small CD since this is the primary training loss, but has difficulty reconstructing the correct point distribution on the partial surface due to the patch-based approach, resulting in much higher EMD for all cases.

**Multiview Reconstruction**: A direct application of TPoint-Net++ is partial shape reconstruction of observed geometry through a union of predicted T-NOCS spatial points. Due to the quantitative accuracy of TPointNet++ at each frame (Tab. 1), aggregated results closely match GT for unseen instances in all categories as shown in Fig. 4 (left).

Table 3: Pose estimation using T-NOCS.

| Method | Category | Trans Err | Rot Err(°) | Point Err |
|---|---|---|---|---|
| RPM-Net | Cars | **0.0049** | **1.1135** | **0.0066** |
| CaSPR | | 0.0077 | 1.3639 | 0.0096 |
| RPM-Net | Chairs | **0.0026** | **0.4601** | **0.0036** |
| CaSPR | | 0.0075 | 1.5035 | 0.0091 |
| RPM-Net | Airplanes | **0.0040** | **0.5931** | **0.0048** |
| CaSPR | | 0.0051 | 0.9456 | 0.0057 |

**Rigid Pose Estimation**: The world–canonical 3D point correspondences from TPointNet++ allow fitting rigid object (or camera) pose at observed frames using RANSAC [21]. Tab. 3 reports median test errors showing TPointNet++ is competitive with RPM-Net [78], a recent **specialized** architecture for robust iterative rigid registration. Note here, RPM-Net takes *both* the raw depth and **GT** T-NOCS points as input. Translation and rotation errors are the distance and degree angle difference from the GT transformation. Point error measures the per frame median distance between the GT T-NOCS points transformed by the predicted pose and the input points. Qualitative results are in Fig. 4 (right).

**Rigid Spatiotemporal Interpolation**: The full CaSPR model can densely sample a sparse input sequence in space-time as shown in Fig. 7. The model takes three input frames of 512 points (corresponding GT T-NOCS points shown on top) and reconstructs an arbitrary number of steps with 2048 points (middle). The representation can be sampled at any ST resolution but, in practice, is limited by memory. The CNF

Table 4: Reconstructing 10 observed timesteps (left) and maintaining temporal correspondences (right) on Warping Cars.

| | *Reconstruction* | | *Correspondences* | |
|---|---|---|---|---|
| **Method** | **CD** | **EMD** | **Dist** $t_1$ | **Dist** $t_{10}$ |
| OFlow | 1.512 | 20.401 | **0.011** | **0.031** |
| CaSPR | **0.955** | **11.530** | 0.013 | 0.035 |

maps Gaussian noise to the visible surface (bottom). Points are most dense in high probability areas (shown in red); in our data this roughly corresponds to where the camera is focused on the object surface at that timestep.

**Non-Rigid Reconstruction and Temporal Correspondences**: CaSPR can represent and reconstruct deformable objects. We evaluate on a variation of the Warping Cars dataset introduced in Occupancy Flow (OFlow) [45] which contains 10-step sequences of *full* point clouds sampled from ShapeNet [5] cars deforming over time. The sequences in this dataset are already consistently aligned and scaled, so CaSPR is trained only using $\mathcal{L}_r$.

Tab. 4 compares CaSPR to OFlow on reconstructing deforming cars at 10 observed time steps (left) and on estimating correspondences over time (right). To measure correspondence error, we (1) sample 2048 points from the representation at $\bar{t}_1$, (2) find their closest points on the GT mesh, and (3) advect the samples to $\bar{t}_{10}$ and measure the mean distance to the corresponding GT points at both steps. Tab. 4 reports median errors over all $\bar{t}_1$ and $\bar{t}_{10}$. For OFlow, samples are advected using the predicted flow field in physical space, while for CaSPR we simply use the same Gaussian samples at each step of the sequence.

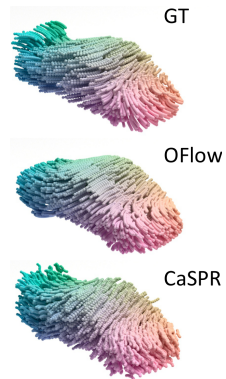

GT

OFlow

CaSPR

Figure 6: Deforming car reconstruction.

CaSPR outperforms OFlow on reconstruction due to overly-smoothed outputs from the occupancy network, while both methods accurately maintain correspondences over time. Note that CaSPR advects system state in a learned latent space and temporal correspondences *naturally emerge* from the CNF when using consistent base samples across timesteps. Fig. 6 visualizes sampled point trajectories for one sequence.

**Cross-Instance Correspondences**:   We observe consistent behavior from the CNF across objects within a shape category too. Fig. 8 shows reconstructed frames from various chair and airplane sequences with points colored by their corresponding location in the sampled Gaussian (before the flow). Similar colors across instances indicate the same part of the base distribution is mapped there. This could potentially be used, for instance, to propagate labels from known to novel object instances.

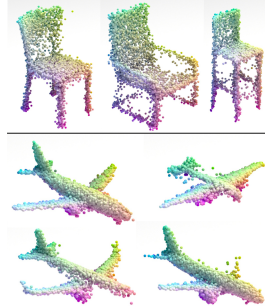

Figure 8: Cross-instance correspondences emerge naturally using a CNF.

**Learning the Arrow of Time**:   A desirable property of ST representations is an understanding of the *unidirectionality* of time [19]: how objects evolve forward in time. We demonstrate this property with CaSPR by training on a dataset of 1000 sequences of a single rigid car where the camera always rotates counter-clockwise at a fixed distance (but random height). CaSPR achieves a median CD of 0.298 and **EMD of 7.114** when reconstructing held-out sequences *forward in time*. However, when the same test sequences are *reversed* by flipping the timestamps, accuracy drastically drops to CD 1.225 and **EMD 88.938**. CaSPR is sensitive to the arrow of time due to the directionality of the Latent ODE and the global temporal view provided by operating on an entire sequence jointly.

**Shape & Motion Disentanglement**   We evaluate how well CaSPR disentangles shape and motion as a result of the latent feature splitting $\mathbf{z}^C = [\mathbf{z}^C_{\text{ST}}, \mathbf{z}^C_{\text{dyn}}]$. For this purpose, we transfer motion between two sequences by embedding both of them using TPointNet++, then taking the static feature $\mathbf{z}^C_{\text{ST}}$ from the first and the dynamic feature $\mathbf{z}^C_{\text{dyn}}$ from the second. Fig. 9 shows qualitative results where each row is a different sequence; the first frame of the shape sequence is on the left, the point trajectories of the motion sequence in the middle, and the final CaSPR-sampled trajectories using the combined feature are on the right. If these features perfectly disentangle shape and motion, we should see the shape of the first sequence with the motion of the second after reconstruction. Apparently, the explicit feature split in CaSPR does disentangle static and dynamic properties of the object to a large extent.

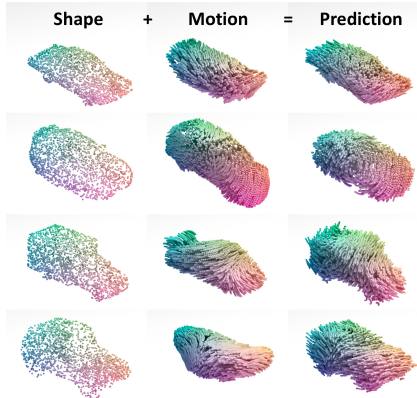

Figure 9: Disentanglement examples on warping cars data.

# 6   Conclusion

We introduced CaSPR, a method to canonicalize and obtain object-centric representations of raw point cloud sequences, which supports spatiotemporal sampling at arbitrary resolutions. We demonstrated CaSPR's utility on rigid and deformable object motion and in applications like spatiotemporal interpolation and estimating correspondences across time and instances.

**Limitations and Future Work**:   CaSPR leaves ample room for future exploration. We currently only support batch processing, but online processing is important for real-time applications. Additionally, CaSPR is expensive to train. Our canonicalization step requires dense supervision of T-NOCS labels which may not be available for real data. While the network is well-suited for ST interpolation, the extrapolation abilities of CaSPR need further investigation. CaSPR is object-centric, and further work is needed to generalize to object collections and scenes. Additionally, outlier shapes can cause noisy sampling results and if the partial view of an object is ambiguous or the object is symmetric, TPointNet++ may predict a flipped or rotated canonical output.

Finally, using a single CNF for spatial sampling is fundamentally limited by an inability to model changes in topology [14, 18]. To capture fine-scale geometric details of shapes, this must be addressed.

## Broader Impact

CaSPR is a fundamental technology allowing the aggregation and propagation of dynamic point cloud information – and as such it has broad applications in areas like autonomous driving, robotics, virtual/augmented reality and medical imaging. We believe that our approach will have a mostly positive impact but we also identify potential undesired consequences below.

Our method will enhance the capabilities of existing sensors and allow us to build models of objects from sparse observations. For instance, in autonomous driving or mixed reality, commonly used LIDAR/depth sensors are limited in terms of spatial and temporal resolution or sampling patterns. Our method creates representations that overcome these limitations due to the capability to continuously sample in space and time. This would enable these sensors to be cheaper and operate at lower spacetime resolutions saving energy and extending hardware lifespans. Our approach could also be useful in spatiotemporal information propagation. We can propagate sparse labels in the input over spacetime, leading to denser supervision. This would save manual human labeling effort.

Like other learning-based methods, CaSPR can produce biased results missing the details in the input. In a self driving scenario, if an input LIDAR point cloud only partially observes a pedestrian, CaSPR may learn representations that misses the pedestrian completely. If real-world systems rely excessively on this incorrect representation it could lead to injuries or fatalities. We look forward to conducting and fostering more research in other applications and negative impacts of our work.

## Acknowledgments and Funding Disclosure

This work was supported by grants from the Stanford-Ford Alliance, the SAIL-Toyota Center for AI Research, the Samsung GRO program, the AWS Machine Learning Awards Program, NSF grant IIS-1763268, and a Vannevar Bush Faculty Fellowship. The authors thank Michael Niemeyer for providing the code and shape models used to generate the warping cars dataset. Toyota Research Institute ("TRI") provided funds to assist the authors with their research but this article solely reflects the opinions and conclusions of its authors and not TRI or any other Toyota entity.

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
