[Supplementary Material · CaSPR_Supplement_CR.pdf]

# Supplementary Document for
# CaSPR: Learning Canonical Spatiotemporal Point Cloud Representations

**Davis Rempe[1]**    **Tolga Birdal[1]**    **Yongheng Zhao[2]**    **Zan Gojcic[3]**
**Srinath Sridhar[1]**    **Leonidas J. Guibas[1]**
[1]Stanford University    [2]University of Padova    [3]ETH Zürich
`geometry.stanford.edu/projects/caspr`

## Abstract

This document provides supporting details, discussion, and results omitted from the main paper due to space constraints. We also urge the reader to view the provided **supplementary video**, which more clearly demonstrates the spatiotemporal capabilities of CaSPR. We expand on discussions in Sec. 1, provide supporting evaluations in Sec. 2, explain details of dataset generation and architecture implementation in Sec. 3 and Sec. 4, and give details of experiments from the main paper in Sec. 5.

## 1 Discussions

**Remarks on ODE-Nets**  The requirements of homeomorphisms and differentiability impose certain limitations. First, neural ODEs lack a universal approximation capability as non-intersecting trajectories cannot learn to approximate arbitrary topologies [21][1]. On the other hand, it is also shown that this very property brings intrinsic robustness to ODE-Nets [8]. Moreover, the requirement of invertibility in CNFs is proven to hamper the approximation quality of the target distribution [4]. In fact, for a perfect recovery and likelihood evaluation, non-invertibility is a requirement [4]. Nonetheless, the extent to which these limitations restrict the applicability of Neural ODEs and CNFs is still an active research topic.

**Why can't we use existing point cloud networks as a canonicalizer?**  Extending PointNet++ to time (similar to MeteorNet [11]) requires some form of a spatiotemporal neighborhood query or using time as an auxiliary input feature diminishing its contribution. Spatiotemporal neighborhood queries are undesirable as they necessitate difficult hyperparameter tuning and limit the network's ability to holistically understand the motion. For example, learning the arrow of time (as CaSPR does in Sec. 5 of the main paper) would be difficult when using local spatiotemporal queries. PointNet can somewhat remedy this by operating on the full 4D point cloud at once, treating time equally important as the spatial dimensions. However, we found that PointNet by itself is incapable of extracting descriptive local features, which are essential for an accurate mapping to T-NOCS.

**On the arrow of time**  Due to the second law of thermodynamics, the entropy of an isolated system tends to increase with time, making the direction of time irreversible [10] *i.e. it is more common for a motion to cause multiple motions than for multiple motions to collapse into one consistent motion* [13]. This causality is confirmed in computer vision by showing that the statistics of natural videos are not symmetric under time reversal [13]. Any method processing spacetime inputs should then be sensitive to this direction so as to yield distinctive representations rather than being *invariant*

Figure 1: Failure cases of CaSPR. The CNF has difficulty capturing local details and very thin structures (left) along with uncommon shapes (middle). TPointNet++ has trouble with symmetry or ambiguity in partial views, resulting in reflected or rotated predictions (right).

to it. As shown in the experiments of the main paper, thanks to the inclusion of timestamps and the Latent ODE advecting forward in time, CaSPR is highly aware of this unidirectionality and it is one of the reasons why it can extract robust spatiotemporal latent features.

**On Disentanglement** In the main paper, we have demonstrated experimentally that static and dynamic feature disentanglement is achieved to a large extent. Note that CaSPR involves no mechanism that can guarantee a theoretically disentangled latent space such as the one of [22]. Our design *softly* encourages the canonicalization network to respect the subspace nature by only advecting the dynamic feature with the ODE. Though this is not a CaSPR-specific drawback and many SoTA disentanglement networks rely upon the same intuition.

**Limitations** Using a CNF to sample the object surface does come with some limitations as mentioned in prior work [19] and discussed above. The inherent properties of CNFs may hamper the ability to capture fine-scale geometric detail. We observe this in chairs with back slats and other thin structures that are not captured by our Reconstruction CNF as shown in the left panel of Fig. 1. Additionally, outlier shapes can cause noisy sampling results (shown in the middle). One current limitation of TPointNet++ is its inability to handle symmetry when canonicalizing a point cloud sequence. If the partial view of an object is ambiguous or the object is symmetric, TPointNet++ may predict a flipped or rotated canonical output as shown in the right panel.

## 2 Additional Evaluations

We provide evaluations of CaSPR omitted from the main paper due to space constraints. Please see Section 5 of the main paper for an explanation of evaluation metrics and the primary results.

Table 1: CaSPR ablations for reconstruction of rigid car sequences over 10 observed frames.

| Method | CD | EMD | NFE |
|---|---|---|---|
| No $\mathcal{L}_c$ | 0.605 | 11.482 | **38.2** |
| No Factorization | 0.635 | 11.249 | 101.3 |
| No Input Aug | 0.577 | 10.253 | 38.9 |
| Full Arch | **0.566** | **10.103** | 39.6 |

### 2.1 Ablation Study

We compare the full CaSPR architecture (*Full Arch*) to multiple ablations in Tab. 1. This includes: (i) not using the canonicalization loss (*No $\mathcal{L}_c$*), (ii) not factorizing the latent ST feature and instead feeding the entire vector to the Latent ODE (*No Factorization*), and (iii) using no pairwise terms (see Sec. 4) as input augmentation (*No Input Aug*). In addition to reconstruction metrics, we report the mean number of function evaluations (NFE) for the Latent ODE. This is the average number of times the ODE solver queries the dynamics network while integrating forward in time for a single sequence. Each method is trained

Table 2: Reconstruction errors with varying numbers of input points per frame for rigid car motion.

| Num Points | CD | EMD |
|---|---|---|
| 2048 | 0.5657 | 10.1028 |
| 1024 | **0.5486** | **10.0339** |
| 512 | 0.5904 | 10.5188 |
| 256 | 0.8222 | 13.8275 |
| 128 | 1.4730 | 20.7233 |

on the rigid cars category and reconstructs all 10 input frames for evaluation. The full CaSPR architecture performs best. Note that the static/dynamic feature factorization is especially important to limit the complexity of Latent ODE dynamics.

## 2.2 Sparsity in Space and Time

We evaluate CaSPR's ability to reconstruct partial point cloud sequences from the rigid car category under sparsity in both space and time. Given 10 input frames, Tab. 2 shows the performance for reconstructing all 10 observed frames with a varying number of points available at each frame. Performance is consistent until 256 or fewer points are given at which point it drops off rapidly. Tab. 3 shows performance when varying the number of available observed timesteps for each test sequence. Observed timesteps are distributed as evenly as possible over the 10-step sequence for this evaluation. Performance is stable even with 3 observed frames, but does significantly drop when only 2 frames are given (*i.e.* the first and last steps).

Table 3: Reconstruction errors with a varying number of observed input frames.

| | Observed | | Unobserved | |
|---|---|---|---|---|
| **Num Observed** | **CD** | **EMD** | **CD** | **EMD** |
| 10 steps | **0.5657** | **10.1028** | — | — |
| 7 steps | 0.5701 | 10.3406 | **0.5609** | **10.0304** |
| 5 steps | 0.5664 | 10.4310 | 0.5620 | 10.3374 |
| 3 steps | 0.5904 | 11.4641 | 0.5837 | 11.2586 |
| 2 steps | 0.7095 | 14.8348 | 0.7233 | 16.3499 |

## 2.3 Reconstructing Longer Sequences

We evaluate CaSPR when trained on the rigid motion car dataset with sequences of 25 frames (rather than 10 as in the main paper). During training, we randomly subsample 10 frames (rather than 5) from each sequence, and evaluate with the full 25-frame sequence as input (rather than 10). Tab. 4 shows reconstruction performance compared to the model in the main paper which uses the 10-frame sequence dataset. We see there is a minimal difference in performance, indicating CaSPR is capable of handling longer-horizon motion.

Table 4: Reconstruction errors for longer sequences on rigid car data.

| **Test Seq Length** | **CD** | **EMD** |
|---|---|---|
| 10 | 0.566 | **10.103** |
| 25 | **0.534** | 10.815 |

## 2.4 Multi-Category Model

We evaluate CaSPR when trained on all shape categories together: cars, chairs, and airplanes. This determines the extent of the category-level restriction on our method. Results compared to models trained on each category separately are shown in Tab. 5. Models are evaluated by reconstructing all 10 observed time steps. As expected, there is a performance drop when training a single joint model, however errors are still reasonable and in most cases better than the *PointFlow* baseline in terms of EMD (see Tab. 2 in main paper).

Table 5: Reconstruction errors training on all categories jointly.

| **Train Data** | **Test Data** | **CD** | **EMD** |
|---|---|---|---|
| Cars | Cars | **0.566** | **10.103** |
| All | Cars | 0.728 | 13.631 |
| Chairs | Chairs | **0.715** | **13.009** |
| All | Chairs | 1.231 | 15.632 |
| Airplanes | Airplanes | **0.231** | **6.026** |
| All | Airplanes | 0.391 | 8.213 |
| All | All | 0.798 | 12.578 |

## 2.5 Canonicalizing for Deformation

We evaluate the ability of TPointNet++ to canonicalize non-rigid transformations. Given a deforming car sequence from the Warping Cars dataset, the task is to remove the deformation at each step, leaving the base shape without any warping. To achieve this, we train TPointNet++ with $\mathcal{L}_c$ only, and supervise every step in a sequence with the same GT canonical point cloud that contains no deformation. Note that Warping Cars is already canonical in terms of rigid transformations, so the network needs to learn to factor out non-rigid deformation only. Results are shown in Tab. 6 where we compare TPointNet++ to a baseline that simply copies the input points to the output (*Identity*, which performs reasonably since there is no rigid transformation). *Identity* trivially gives a perfect time error, but TPointNet++ achieves a much lower spatial error, effectively removing the deformation from each step. This is qualitatively shown in Fig. 10. This strategy of canonicalization offers an explicit way to extract temporal correspondences over time, rather than relying on the CNF to naturally exhibit correspondences (main paper Sec. 5).

Table 6: Canonicalization performance for deforming cars.

| **Method** | **Spatial Err** | **Time Err** |
|---|---|---|
| Identity | 0.0583 | **0.0000** |
| TPointNet++ | **0.0221** | 0.0012 |

## 2.6 Label Propagation through Canonicalization

We evaluate the ability of T-NOCS canonicalization to establish correspondences by propagating point-wise labels both throughout a sequence and to new sequences of different object instances. Given a semantic segmentation of the partial point cloud at the *first* frame of a sequence at time $s_1$,

Figure 2: Example of semantic segmentation label propagation over time and across instances through T-NOCS canonicalization. The given labels in the first frame of the top sequence (orange box) are transferred to later frames in the same sequence (green dashed box) and to other sequences with different object instances (blue dashed boxes) by comparing to the labeled frame in the shared canonical space.

the first task is to label all subsequent steps in the sequence at times $s_2, \ldots, s_k$, *i.e.* propagate the segmentation forward in time. Secondly, we want to label all frames of sequences containing *different* object instances *i.e.* propagate the segmentation to different objects of the same class. We achieve both through canonicalization with TPointNet++: all frames in each sequence are mapped to T-NOCS, then unknown points are labeled by finding the closest point in the given labeled frame at $s_1$. If the closest point in $s_1$ is not within a distance of $0.05$ in the canonical space, it is marked "Unknown". This may happen if part of the shape is not visible in the first frame due to self-occlusions.

Results of this label propagation for a subset of the chairs (1315 sequences) and airplanes (1215 sequences) categories of the rigid motion test set are shown in Tab. 7. We report median point-wise accuracy over all points (*Total Acc*) and for points successfully labeled by our approach (*Known Acc*). For the instance propagation task, we randomly use $1/3$ of test sequences as "source" sequences where the first frame is labeled, and the other $2/3$ are "target" sequences to which labels are propagated. In this case, accuracy is reported only for target sequences. Qualitative results are shown in Fig. 2.

Table 7: Segmentation label propagation performance. *Total Acc* is point-wise accuracy over all points; *Known Acc* is only for points that our method successfully labels.

| Task | Category | Total Acc | Known Acc |
|------|----------|-----------|-----------|
| Temporal Propagation | Chairs | 0.9419 | 0.9804 |
| | Airplanes | 0.9580 | 0.9676 |
| Instance Propagation | Chairs | 0.6553 | 0.8425 |
| | Airplanes | 0.7744 | 0.8006 |

## 2.7 Extrapolating Motion

We evaluate CaSPR's ability to extrapolate future motion without being explicitly trained to do so. In particular, the model is given the first 5 frames in each sequence and must predict the following 5 frames. The ability to predict future motion based on the learned prior would be valuable in real-time settings. We evaluate the already-trained full reconstruction models for each object category (from Tab. 2 in the main paper). Note that these models are supervised with observed frames - they are not trained to predict unseen future states. Results are shown in Tab. 8. Clearly there is a sharp performance drop between observed and extrapolated frames as we might expect, though performance is actually on par with the AtlasNet baseline (Tab. 2, main paper) in some cases. We note that qualitatively, the model produces reasonable future motion

Table 8: Reconstruction of extrapolated frames.

| Category | 5 Observed | | 5 Extrapolated | |
|----------|-----------|-----|---------------|-----|
| | CD | EMD | CD | EMD |
| Cars | 0.597 | 9.833 | 1.023 | 21.055 |
| Chairs | 0.687 | 12.502 | 1.010 | 20.648 |
| Airplanes | 0.224 | 5.719 | 0.286 | 9.625 |

Figure 4: Examples from the rigid motion dataset. Partial point cloud sequences resulting from rendered data depth maps are shown; color shifts from blue to red over time.

based on what it has seen and even hallucinates unseen parts of the shape, though it cannot handle sudden changes in direction.

# 3 Datasets Details

**Rigid Motion Dataset**    Please see Section 5 of the main paper for an introduction to our new dataset containing rigid motion for ShapeNet [2] cars, chairs, and airplanes. This simulated dataset gives us the ability to capture a wide range of trajectories and acquire the necessary inputs and supervision to train and evaluate CaSPR.

We generate these motions within the Unity game engine[2]. For each object instance, we simulate a camera trajectory around the object (placed at the origin) that starts at a random location and continues for 50 timesteps. The camera always points towards the origin and its location is parameterized as a point on the surface of a sphere centered at the origin: by a longitudinal and latitudinal angle along with a radius. To produce a trajectory, each of these parameters is gradually increased or decreased independently. When a parameter reaches a set limit, its direction is reversed, producing interesting and challenging motions. At each step of the trajectory, a depth map and NOCS map [17] are rendered from the current camera view. An example NOCS map from the dataset is shown in Fig. 3. Example camera trajectories and the resulting aggregate canonical point cloud are shown in Fig. 8.

Figure 3: NOCS map from rigid motion dataset.

The rendered frames are further processed to produce the final dataset of raw depth and canonical T-NOCS point cloud sequences. The rendered trajectory for each object instance is split into 5 sequences (with 10 steps each). 4096 pixels on the object are uniformly sampled from each depth map to extract *raw* partial point cloud sequences in the world (camera) frame that are used as the input to CaSPR. Examples of these partial sequences are shown in Fig. 4. Each input point cloud in a sequence is given a timestamp in uniform steps from 0.0 to 5.0. The same sampled pixels are taken from the NOCS map to extract a corresponding canonical partial point cloud and given a timestamp from 0.0 to 1.0: this represents the supervision for CaSPR. In total, the car category contains 2527 object instances (12, 635 sequences), chairs contains 5000 objects (25, 000 sequences), and airplanes has 4045 objects (20, 225 sequences). Each category is split 80/10/10 into train/val/test. The val/test sets are entirely made up of object instances and camera motions that do not appear in the training split.

Note that during training and inference, only a subset of the available 4096 points at each step in the dataset are used, as detailed in the main paper (usually 1024 during training and 2048 during evaluation). Additionally, during training a subset of the available 10 frames are randomly sampled from each sequence, giving non-uniform step sizes between observations. These subsampled sequences are shifted so that $s_1 = 0.0$ before being given to CaSPR, making things practically easier as it ensures that the Latent ODE always starts from $\bar{t}_1 = 0$ for any sequence in a batch.

**Warping Cars Dataset**   In Section 5.1 of the main paper ("Non-Rigid Reconstruction and Temporal Correspondences"), we use a variation of the Warping Cars dataset from Occupancy Flow (OFlow) [12]. We generate our version of this dataset with code kindly provided by the authors of that work. The dataset contains the same car models as our rigid motion dataset, however they are watertight versions that allow determining occupancy, which is needed to train OFlow. Same as the rigid motion dataset, we generate 5 sequences for each car instance with 10 frames of motion each. Consistent with the OFlow paper, we sample 100k points per sequence on the surface of the object that are in correspondence over time and can be used as inputs to CaSPR and OFlow; we also sample 100k points in the unit cube containing the object with corresponding occupancy labels for OFlow. Note that this data gives point clouds on the *complete* object rather than the partial surface, and there is no rigid motion in the dataset – only deformation. This means the sequences are already canonical in the sense that cars are consistently aligned and scaled. We also use input timestamps from 0.0 to 1.0, so the data is already canonical in time as well.

# 4   Implementation Details

We next cover additional architectural and training details of our method. Please see Section 4.1 of the main paper for the primary discussion of our architecture and training procedure. We implement our method using PyTorch[3].

**TPointNet++**   The PointNet [14] component operates on the entire 4D input point cloud and extracts a 1024-dimensional global feature and 64-dimensional per-point features. We use the vanilla classification PointNet architecture with 3 shared fully-connected (FC) layers (64, 128, 1024), ReLU non-linearities, and a final max-pool function. The per-point features come from the output of the first FC layer, while the global feature is the output of the max-pool. We do not use the input or feature transform layers, and replace all batch normalization with group normalization [18] using 16 groups, which is crucial to good performance with small batch sizes.

The PointNet++ [15] component operates on each frame of the point cloud sequence independently and does not receive the timestamp as input. The input points to this part of the network are augmented with pairwise terms $x^2$, $y^2$, $z^2$, $xy$, $yz$, and $xz$, which we found improves reconstruction performance (see Sec. 2.1). We use a modified version of the segmentation architecture which contains 5 set abstraction (SA) layers (PointNet dimensions, radii, number points out): $([[16, 16, 32], [32, 32, 64]], [0.8, 0.4], 1024) \rightarrow ([[32, 32, 64], [32, 32, 64]], [0.4, 0.2], 512) \rightarrow ([[64, 64, 128], [64, 96, 128]], [0.2, 0.1], 256) \rightarrow ([[128, 256, 256], [128, 256, 256]], [0.1, 0.05], 64) \rightarrow ([[256, 256, 512], [256, 256, 512]], [0.05, 0.02], 16)$. These are followed by 5 feature propagation (FP) layers which each have 2 layers with hidden size $512$, and a final shared MLP with layers $(512, 512)$ to produce the final per-point 512-dimensional local feature. ReLU non-linearities are used throughout, and we again replace all batch normalization with group normalization [18] using 16 groups.

The final shared MLP which processes the concatenated features from PointNet and PointNet++ also uses group normalization and ReLU.

There are a few things of note with this architecture. First of all, it avoids any spatiotemporal neighborhood queries since time is handled entirely with PointNet which treats the timestamps as an additional spatial dimension. This allows the network to decide which time windows are most important to focus on. Second, the architecture can easily generalize to sequences with differing numbers of points and frames since both are processed almost entirely independently (the only components affected by changing these at test-time are the PointNet max-pooling and the PointNet++ spatial neighborhood queries).

**Latent ODE**   The Latent ODE is given a 64-dimensional latent state $\mathbf{z}_0 \triangleq \mathbf{z}_{\text{dyn}}^C$ which can be advected to any canonical timestamp from 0.0 to 1.0. The dynamics of the Latent ODE is an MLP with 3 hidden layers (512, 512, 512) which uses Tanh non-linearities. We use the *torchdiffeq* package[4] [3] which implements both the ODE solver along with the adjoint method to enable

backpropagation. We use the *dopri15* solver which is an adaptive-step Runge-Kutta 4(5) method. We use a relative tolerance of 1e-3 and absolute tolerance of 1e-4 both at training and test time.

**Reconstruction CNF**   Our reconstruction CNF adapts the implementation of FFJORD [6] for point clouds from PointFlow [19]. The dynamics of the CNF are parameterized by a neural network that uses 3 hidden *ConcatSquashLinear* layers (512, 512, 512), which are preceeded and followed by a *Moving Batch Normalization* layer. We use Softplus non-linearities after each layer. Please see [19] for full details. In short, each layer takes as input the current hidden state (512-dimensional at hidden layers or 3-dimensional $x, y, z$ at the first layer), the conditioning shape feature (1600-dimensional in CaSPR), and the current time of the flow (scalar), and uses this information to update the hidden state (or output the 3-dimensional derivative at the last layer). The ODE is again solved using *dopri15*, this time with both a relative and absolute tolerence of 1e-5. We use the adjoint method for backpropagation and jointly optimize for the final flow time $T$ along with the parameters of network.

**Training and Inference**   In practice, the full loss function is $\mathcal{L} = w_r \mathcal{L}_r + w_c \mathcal{L}_c$ where the contributions of the reconstruction and canonicalization terms are weighted as $w_r = 0.01$ and $w_c = 100$ as to be similar scales. No weight decay is used. We use the Adam [9] optimizer ($\beta_1 = 0.9$, $\beta_2 = 0.999$) with a learning rate of 1e-4. During training, we periodically compute the validation set loss, and after convergence use the weights with the best validation performance as the final trained model. The number of epochs trained depends on the dataset and the task. We train across up to 4 NVIDIA Tesla V100 GPUs which allows for a batch size of up to 20 sequences of 5 frames each. As noted in previous work [19], solving and backpropagating through ODEs (two in our case: Latent and CNF) results in slow training: it takes about 5 days for the full CaSPR architecture using the multi-gpu setup. The full CaSPR network contains about 16 million trainable parameters. Inference for a 10-step sequence of rigid car motion with 2048 points at each step takes on average 0.598 seconds.

## 5   Experimental Details and Supplemental Results

Here we give details of experiments shown in Section 5 of the main paper along with some supporting results for these experiments (*e.g.* means, standard deviations, and visualizations).

**Evaluation Procedure**   To evaluate reconstruction error, we use the Chamfer Distance (CD) and Earth Mover's Distance (EMD). For our purposes, we define the CD between two point clouds $\mathcal{X}_1, \mathcal{X}_2$ each with $N$ points as

$$d_{CD}\left(\mathcal{X}_1, \mathcal{X}_2\right) = \frac{1}{N} \sum_{\mathbf{x}_1 \in \mathcal{X}_1} \min_{\mathbf{x}_2 \in \mathcal{X}_2} \|\mathbf{x}_1 - \mathbf{x}_2\|_2^2 + \frac{1}{N} \sum_{\mathbf{x}_2 \in \mathcal{X}_2} \min_{\mathbf{x}_1 \in \mathcal{X}_1} \|\mathbf{x}_1 - \mathbf{x}_2\|_2^2$$

and the EMD as

$$d_{EMD}\left(\mathcal{X}_1, \mathcal{X}_2\right) = \min_{\phi:\mathcal{X}_1 \to \mathcal{X}_2} \frac{1}{N} \sum_{\mathbf{x}_1 \in \mathcal{X}_1} \|\mathbf{x}_1 - \phi(\mathbf{x}_1)\|_2^2$$

where $\phi : \mathcal{X}_1 \to \mathcal{X}_2$ is a bijection. In practice, we use a fast approximation of the EMD based on [1]. Both CD and EMD are always reported multiplied by $10^3$.

As noted in the main paper, for these reconstruction metrics and the canonicalization error metrics, we report the median values over all test frames. This is motivated by the fact that ShapeNet [2] contains some outlier shapes which result in large errors that unfairly bias the mean and do not accurately reflect comprehensive method performance. For completeness, we also report mean and standard deviation for these metrics in this document for main paper experiments. Note that CD and EMD, along with the spatial canonicalization error, are all

Table 9: Canonicalization performance mean and (standard deviation). Supplements Tab. 1 in the main paper.

| Method | Category | Spatial Err | Time Err |
|---|---|---|---|
| MeteorNet | Cars | 0.0834 (0.0801) | **0.0002** (0.0015) |
| PointNet++ No Time | | 0.0649 (0.0468) | — |
| PointNet++ w/ Time | | 0.0715 (0.0811) | 0.0006 (0.0012) |
| PointNet | | 0.0485 (0.0952) | 0.0016 (0.0015) |
| TPointNet++ No Aug | | 0.0225 (0.0501) | 0.0015 (0.0014) |
| TPointNet++ No Time | | **0.0224** (0.0570) | — |
| TPointNet++ | Cars | 0.0229 (0.0617) | 0.0013 (0.0012) |
| TPointNet++ | Chairs | 0.0162 (0.0337) | 0.0008 (0.0006) |
| TPointNet++ | Airplanes | 0.0148 (0.0412) | 0.0009 (0.0007) |

reported in the canonical space where the shape lies within a unit cube. This helps intuit the severity of reported errors.

Although we randomly subsample 1024 points at each frame for training, during evaluation we always use the same 2048 points (unless specifically stated otherwise) to make evaluation consistent across compared methods. Unless otherwise stated, CaSPR and all compared baselines reconstruct the same number of points as in the input (*e.g.* for evaluation, each input frame has 2048 points, so we sample 2048 points from our Reconstruction CNF).

**Canonicalization** In this experiment, we train TPointNet++ by itself with only the canonicalization loss $\mathcal{L}_c$ on each category of the rigid motion dataset. In order to make the number of parameters comparable across all baselines, we use hidden layers of size 1024 (rather than 1600) in the final shared MLP for the full TPointNet++ architecture only. We compare to the following baselines which are all trained with the same $\mathcal{L}_c$:

- *MeteorNet* [11]: A recent method that extends PointNet++ to process point cloud sequences through spatiotemporal neighborhood queries. We adapt the *MeteorNet-seg* version of the architecture with *direct grouping* for our task by adding an additional *meteor direct module* layer, as well as two fully connected layers before the output layer. Additionally, we slightly modify feature sizes to make the model capacity comparable to other methods. We found the spatiotemporal radii hyperparameters difficult to tune and in the end we opted for 10 uniformly sampled radii between $(0.03, 0.05)$ in the first layer, which were doubled in each subsequent layer.

- *PointNet++ No Time*: An ablation of TPointNet++ that removes the PointNet component. This leaves PointNet++ processing each frame independently followed by the shared MLP, and therefore has no notion of time.

- *PointNet++ w/ Time*: This is the same ablation as above, but modified so that the PointNet++ receives the timestamp of each point as an additional input feature. Note that local neighborhood queries are still performed only on spatial points, but they may be across timesteps so we use increased radii of $(0.05, 0.1, 0.2, 0.6, 1.2, 2.0)$. This baseline represents a naive way to incorporate time, but dilutes its contributions since it is only an auxiliary feature.

- *PointNet*: An ablation of TPointNet++ that removes the PointNet++ component. This leaves only PointNet operating on the full 4D spatiotemporal point cloud. This baseline treats time equally as the spatial dimensions, but inherently lacks local geometric features.

- *TPointNet++ No Time*: An ablation of TPointNet++ that only regresses the spatial part of the T-NOCS coordinate (and not the normalized timestamp). This baseline still takes the timestamps as input, it just doesn't regress the last time coordinate.

- *TPointNet++ No Aug*: An ablation of TPointNet++ that does not augment the input points to PointNet++ with pairwise terms as described previously. This baseline was omitted from the main paper for brevity, so a comparison of median performance is shown in Tab. 10.

Each model is trained for 220 epochs on the cars category. TPointNet++ is trained for 120 and 70 epochs on the airplanes and chairs categories, respectively, due to the increased number of objects. Median canonicalization errors are in Tab. 1 of the main paper; the mean and standard deviations are shown in Tab. 9.

Table 10: Canonicalization performance without input augmentation.

| Method | Category | Spatial Err | Time Err |
|---|---|---|---|
| No Aug | Cars | 0.0138 | 0.0012 |
| Full Arch | Cars | **0.0118** | **0.0011** |

**Representation and Reconstruction** In this experiment, we compare the full CaSPR architecture to two baselines on the task of reconstructing a partial point cloud sequence.

The baselines represent one alternative to achieve spatial continuity, and one to achieve temporal continuity. The *CaSPR-Atlas* baseline is the full CaSPR architecture as described, but replaces the Reconstruction CNF with an AtlasNet [7] decoder. We use the same decoder as the original AtlasNet. This decoder contains 64 MLPs, each responsible for transforming a patch to the partial visible surface at a desired timestep. Each MLP contains 4 hidden layers $(1600, 1600, 800, 400)$ with Tanh activation functions. This version of CaSPR is still trained with the auxiliary canonicalization task ($\mathcal{L}_c$ loss), but the reconstruction loss is now a Chamfer distance since AtlasNet does not support

Table 11: Partial surface sequence reconstruction results showing mean and (standard deviation). Supplements Tab. 2 in the main paper.

| Method | Category | 10 Observed | | 3 Observed | | 7 Unobserved | |
|---|---|---|---|---|---|---|---|
| | | CD | EMD | CD | EMD | CD | EMD |
| PointFlow | Cars | **0.537** (0.272) | 15.986 (11.130) | **0.538** (0.270) | **15.967** (11.065) | **0.700** (0.732) | 17.362 (12.276) |
| CaSPR-Atlas | Cars | 0.814 (1.729) | 26.922 (28.562) | 0.874 (2.051) | 29.171 (29.479) | 0.853 (1.705) | 26.416 (27.582) |
| CaSPR | Cars | 0.795 (1.048) | **14.242** (21.619) | 0.846 (1.261) | 16.564 (24.296) | 0.824 (1.108) | **16.217** (23.011) |
| PointFlow | Chairs | **0.907** (0.519) | 20.254 (11.938) | **0.907** (0.514) | 20.225 (11.899) | 1.245 (1.299) | 21.971 (13.417) |
| CaSPR-Atlas | Chairs | 1.007 (1.243) | 54.406 (24.970) | 1.030 (1.221) | 54.827 (25.250) | 1.061 (1.277) | 52.964 (24.355) |
| CaSPR | Chairs | 1.013 (1.426) | **15.287** (9.837) | 0.972 (1.498) | **15.757** (11.154) | **1.000** (1.542) | **16.145** (11.620) |
| PointFlow | Airplanes | **0.367** (0.366) | 11.852 (8.768) | **0.366** (0.363) | 11.862 (8.725) | **0.446** (0.527) | 12.335 (9.146) |
| CaSPR-Atlas | Airplanes | 0.587 (1.196) | 23.444 (17.386) | 0.653 (1.369) | 23.165 (16.932) | 0.663 (1.400) | 22.661 (16.853) |
| CaSPR | Airplanes | 0.536 (1.468) | **8.827** (12.650) | 0.536 (1.682) | **8.992** (13.219) | 0.530 (1.673) | **9.031** (12.792) |

Figure 5: Reconstruction performance of the *CaSPR-Atlas* and *PointFlow* baselines compared to the full CaSPR model. Each row shows a frame from a different 10-step rigid motion sequence.

likelihood evaluations like a CNF. We use group normalization [18] instead of batch normalization within the decoder to improve performance with small batch sizes.

The *PointFlow* [19] baseline uses their deterministic autoencoder architecture. This follows the autoencoding evaluations from the original paper and uses a PointNet-like encoder to extract a shape feature, which conditions a CNF decoder. This version of the model is trained only with the reconstruction likelihood objective from the CNF, and does not use the various losses associated with the VAE formulation of their architecture. To make it a fair comparison, we increase the size of the shape feature bottleneck to 1600. The CNF decoder uses a dynamics MLP with 3 hidden layers of size $(512, 512, 512)$, just like CaSPR. Also like CaSPR, we train *PointFlow* with a learning rate of 1e-4, which we found to decrease the complexity of dynamics and therefore training time.

Figure 6: Canonicalization, aggregation, and dense reconstruction of rigid motion sequences by the full CaSPR model. Each sequence shows (a) the 10 observed raw partial point cloud frames given as input to CaSPR, (b) the GT partial reconstruction based on the observed frames, (c) the partial reconstruction achieved by aggregating T-NOCS predictions from TPointNet++ with color mapped to spatial error, (d) the aggregated prediction after reconstructing the 10 observed frames with the CNF, and (e) the aggregated prediction when interpolating 30 frames using the CNF.

The *PointFlow* baseline operates on **single already-canonical partial point cloud frames**, while CaSPR and *CaSPR-Atlas* take in raw world-space sequences of partial point clouds. To reconstruct a sequence, *PointFlow* can easily reconstruct the observed (canonical) frames by simply autoencoding each frame independently. However, to allow reconstruction of intermediate unobserved steps, we must use linear interpolation in the shape feature space from surrounding observed frames, as described in the main paper.

Median reconstruction errors are presented in Tab. 2 of the main paper. Mean and standard deviation are shown here in Tab. 11. Generally, the CaSPR variants have a higher standard deviation than PointFlow. This is likely because CaSPR methods must canonicalize the input in addition to reconstructing it, so any errors in this first step may compound in the reconstruction causing some occasional high errors. A qualitative comparison is shown in Fig. 5. The *CaSPR-Atlas* baseline has perhaps deceivingly poor EMD errors. As discussed in the main paper, the patch-based approach has difficulty reconstructing the true point distribution of the partial view and may cause some areas to be much more dense or sparse than they should (see chairs in Fig. 5). Because EMD requires a bijection,

Figure 7: Examples of spatiotemporal interpolation to reconstruct sparse, partial input sequences. The sparse GT canonical point cloud for each sequence is shown in the top row; the dense CaSPR reconstruction using the CNF is shown in the bottom row.

these overly dense areas are paired with distant points causing large errors. However, qualitative and CD results suggest the approach has some advantages: the reconstructed point cloud tends to be less noisy and capture local detail better than its CNF-based counterparts.

Additional results of the full CaSPR model reconstructing 10-frame input sequences of rigid, partial point clouds are shown in Fig. 6. Please see the caption for details. Note that the shown T-NOCS predictions are using TPointNet++ trained jointly within the full CaSPR model rather than individually as in the "Canonicalization" experiments.

**Rigid Spatiotemporal Interpolation**    Additional results of the full CaSPR architecture reconstructing a sparse, partial input sequence are shown in Fig. 7. In each sequence, the model is given 3 frames with 512 points (with GT canonical point cloud shown as *Sparse GT*) and reconstructs any number of densely sampled steps (10 are shown as *Dense Sampling*, each with 2048 points).

**Rigid Pose Estimation**    We solve for object pose in a post-processing step that leverages the world-canonical correspondences given by the output of TPointNet++. We use the full TPoint-Net++ architecture trained as in the "Canoni-calization" evaluation above. For each frame independently, we run RANSAC [5] using 4 points to perform the fitting and with an inlier threshold of 0.015.

Table 12: Pose estimation performance showing mean and (standard deviation). Supplements Tab. 3 in the main paper.

| Method | Category | Trans Err | Rot Err(°) | Point Err |
|---|---|---|---|---|
| RPM-Net | Cars | **0.0071** (0.0102) | **2.1677** (10.0952) | **0.0087** (0.0146) |
| CaSPR | | 0.0116 (0.0245) | 4.9597 (22.8311) | 0.0203 (0.0645) |
| RPM-Net | Chairs | **0.0029** (0.0031) | **0.6212** (3.3530) | **0.0042** (0.0078) |
| CaSPR | | 0.0094 (0.0127) | 3.0264 (9.5897) | 0.0152 (0.0367) |
| RPM-Net | Airplanes | **0.0050** (0.0076) | **2.2703** (16.2945) | **0.0070** (0.0190) |
| CaSPR | | 0.0083 (0.0144) | 3.6740 (16.9152) | 0.0144 (0.0456) |

We compare our approach to a recent method for robust rigid registration called RPM-Net [20]. This is an algorithm specially designed for pairwise point cloud registration that iteratively estimates the transformation parameters of possibly-partial point clouds by estimating soft correspondences in the inferred feature space. Because this method can only operate on pairs of point clouds, during training we give it the raw partial point cloud (1024 points) along with the corresponding GT canonical point cloud (1024 points with permuted ordering) as input. This contrasts with TPointNet++ that only receives the raw partial point cloud at each step, and must *predict* the canonical point cloud to establish correspondences. At test-time, we instead use 2048 points for the input point clouds, and the raw points and GT canonical points are randomly sampled so they are not in perfect correspondence.

Median errors appear in Tab. 3 of the main paper, but mean and standard deviation results are shown here in Tab. 12. Though TPointNet++ does not outperform RPM-Net on any shape category, the minimal gap in performance is impressive considering that TPointNet++ has to solve a much harder

Figure 8: Additional camera pose estimation results. Ground truth trajectories are shown in solid green and the CaSPR prediction in dashed red.

Figure 9: Rigid pose estimation comparison. Each column shows a frame from a different partial point cloud sequence. Predicted object pose (red points) is shown compared to the GT depth point cloud (green points) for each method. Both methods are very accurate.

task (*pose estimation*) than RPM-Net, which receives both the world and canonical point clouds as input and iteratively solves the simpler *pairwise registration* task. As seen in Fig. 9, the qualitative difference between the two methods is nearly imperceivable. Additional qualitative camera pose estimation results from CaSPR are shown in Fig. 8.

**Non-Rigid Reconstruction and Temporal Correspondences**  We compare CaSPR to Occupancy Flow (OFlow) [12] on the task of reconstructing Warping Cars sequences and estimating correspondences over time. Because OFlow uses an implicit occupancy shape representation, this dataset contains complete shapes with a clearly defined inside and out. The OFlow baseline uses the point cloud completion version of the model, which leverages a PointNet-ResNet architecture for both the spatial and temporal encoders. OFlow is trained with the reconstruction loss only (*i.e.* it does not explicitly use a correspondence loss).

Both methods are trained on sequences of 10 frames with 512 points, and tested on sequences of 10 frames with 2048 points. Due to restrictions of the OFlow encoder, the points at each frame in the input sequence are

Table 13: Reconstruction and correspondences mean and (standard deviation) on Warping Cars. Supplements Tab. 4 in the main paper.

| Method | Reconstruction | | Correspondences | |
|---|---|---|---|---|
| | **CD** | **EMD** | **Dist** $t_1$ | **Dist** $t_{10}$ |
| OFlow | 1.764 (0.913) | 24.247 (14.74) | **0.011** (0.003) | **0.032** (0.007) |
| CaSPR | **0.992** (0.256) | **12.864** (5.856) | 0.014 (0.002) | 0.037 (0.009) |

in correspondence over time (note that this is **not** a requirement for CaSPR, which can accurately estimate temporal correspondence even if this is not the case as in most real-world applications) and we must use the same number of timesteps at training and test time. To reconstruct a sequence with OFlow, we reconstruct the mesh at the first time step based on the occupancy network predictions, then randomly sample 2048 points on this mesh and advect them forward in time with the predicted flow field. For CaSPR, we advect the latent feature forward in time to each desired timestep as per the usual, then reconstruct each frame with the CNF using the *same* Gaussian samples at each step to achieve temporal continuity.

Figure 10: Reconstruction results on Warping Cars data. Each sequence is 10 steps in length and we show point trajectories over time for (a) the ground truth input sequence, (b) the reconstruction from Occupancy Flow, (c) the reconstruction at the 10 observed steps with CaSPR, (d) 30 interpolated steps with CaSPR, and (e) the T-NOCS prediction from TPointNet++.

Median reconstruction and correspondence errors are reported in Tab. 4 of the main paper. Here we show the mean and standard deviations in Tab. 13. Reconstruction errors are measured at all 10 observed timesteps by randomly sampling 2048 ground truth points, while correspondence errors are measured at only the first and last steps using the procedure detailed in the main paper. Additional qualitative results are shown in Fig. 10.

## Footnotes

[1]Augmented-Neural ODEs [16] propose to operate on a higher dimensional space as one workaround.

[2]https://unity.com/

[3]https://pytorch.org/

[4]https://github.com/rtqichen/torchdiffeq