[Reviews · NeurIPS 2020]

Review 1

Summary and Contributions: This work presents CaSPR a novel system to represent spatial-temporal sequences of 3D pointclouds. The proposed method allows intra-class generalization to unseen geometry and poses. Unlike previous methods, CaSPR treats time explicitly rather than as another spatial dimension. CaSPR works by learning an embedding from a point cloud sequence into a normalized space (NOCS), then learning a Neural ODE over latent vectors, such that the geometry generated by pushing those latent vectors through a generative model agrees with the normalized geometry in NOCS space. The neural ODE ensures continuity in time respecting temporal unconventionality, while the generative model (CNF) ensures continuity in space.

Strengths: The proposed formulation is elegant and well designed for the task of modelling temporal sequences of deforming objects. The paper is extremely clear, well written and easy to understand. The experiments section does a great job at highlighting the effects of individual design choices (e.g. TPointNet++ vs other architectures for canonicalization). Furthermore, the authors demonstrate the flexibility of their model on a number of downstream reconstruction tasks, showing that it is at least competitive with specialized architectures for these tasks. I also appreciate how the authors reported EMD in addition to Chamfer distances, as I find it a more meaningful metric that is left out of many point-cloud reconstruction papers.

Weaknesses: As the authors mentioned, the method does require a fairly onerous data setup to train (supervised NOCS) which can be difficult to acquire. While this is a limitation, solving this problem is beyond the scope of the paper. The authors also mention that the method is object centric, yet I believe that solving the problem of representing continuous dynamics of objects is a hard problem on its own and extending this problem to full scenes is beyond the scope of this work.

Correctness: Yes as far as I can tell.

Clarity: Yes, very.

Relation to Prior Work: This paper did a great job positioning itself with respect to prior work.

Reproducibility: Yes

Additional Feedback: This is an obvious accept from me, great work! ------- After reading the rebuttal and other reviews, I still feel like this paper is a clear accept.


Review 2

Summary and Contributions: This paper aims at learning object-level representations that aggregate and encode spatiotemporal changes in shapes observed from the 3D sensors. It first proposed an encoder network to canonicalize the point cloud sequence, then exploit latent NeuralODE and CNF to generate novel shapes in spacetime. This work has a wide range of potential applications.

Strengths: This work is a solid one. The proposed problem setup is a novel and useful one, which essentially aims at seeking a unified representation to describe a partially observed point cloud sequence. On the other hand, to resolve the proposed problem, the author(s) exploits and improve several existing works, e.g., NOCS, NeuralODE, to address the problem. This work also demonstrates several potential applications of the proposal. This work is interesting and can benefit the community.

Weaknesses: I have quite a few concerns over this work: 1. As mentioned by the authors, this proposal is entirely based on object-level point cloud sequence, it will be very beneficial to generalize this methodology to scene-level point clouds. 2. The canonicalization network training requires the ground-truth for supervision, however, in practice it is difficult to obtain such ground-truth (as also mentioned by the author(s)), is it possible to perform this canonicalization network training with some unsupervised criterion? 3. The network training needs to be performed for the individual category which is not practical in the real world, it will be useful to make the network to be able to handle different categories of objects at the same time.

Correctness: Basically correct.

Clarity: This paper is well written.

Relation to Prior Work: Yes, this work have differentiate its problem settings with existing works.

Reproducibility: Yes

Additional Feedback: 1. In Section 3, the authors split the latent representation into a static descriptor and a dynamic descriptor. However, why this splitting is reasonable? From the perspective of the canonicalization network, there is no mechanism in the architecture that can guarantee such splitting is achievable. 2. How can the proposed CaSPR guarantee the generated flow (e.g., Figure 7) is scene flow (which represents the actual movements of the particles) rather than a deformation flow (which only reshapes the point cloud to make it have a similar shape to the target point cloud). 3. To generate point cloud sequence with temporal correspondence, does the Gaussian noise needs to be the same one throughout time? Having read the authors's responses and the other reviewers' comments, I think the idea of this paper is good but not surprisingly new. The CNF decoder (generator) deployed at the very end of the model can be replaced by other types of decoder (e.g., AtlasNet decoder), and the latent ODE part can also be replaced by another network module taking time stamp t and initial codeword/feature as input and generate a new feature. In a sense, the authors have put several existing things together to achieve their goals. Having said that, I still think this is an interesting combination and can be helpful for the NeuIPS community. Acceptance to this work is recommended.


Review 3

Summary and Contributions: The authors propose CaSPR, which is a method to learn object-centric Canonical Spatio Temporal Point Cloud Representations of dynamically moving or evolving objects. The CaSPR learns representations that support spacetime continuity, are robust to variable and irregularly spacetime-sampled point clouds, and generalize to unseen object instances. Experimental results demonstrate the effectiveness of the proposed CaSPR on several applications including shape reconstruction, camera pose estimation, continuous spatiotemporal sequence reconstruction, and correspondence estimation from irregularly or intermittently sampled observations.

Strengths: 1. The proposed CaSPR is novel in the following two ways: a) canonicalizing an input point cloud sequence (partial or complete) into a shared 4D container space; b) learning a continuous ST latent representation of top of the canonicalized space. 2. The proposed model can be used in many applications, including partial or full shape reconstruction, spatiotemporal sequence recovery, camera pose estimation, and correspondence estimation. Hence, the proposed method could have a strong impact on the research community.

Weaknesses: In the experiments section, the authors introduce many applications using CaSPR. However, It would be better if the authors could give more details on how CaSPR is adopted to different applications.

Correctness: The method is correct to my best knowledge. Open for discussion.

Clarity: This paper is a completely finished work. The writing is clear.

Relation to Prior Work: The related work is reasonably good. However, it would be better if the authors could put more related work into the main paper instead of supplementary material.

Reproducibility: Yes

Additional Feedback:


Review 4

Summary and Contributions: This paper proposes CaSPR, a framework to reason spatiotemporally-canonicalized object space for point clouds via neural ordinary differential equations and continuous normalizing flows. Experiments demonstrate the effectiveness of the proposed method on various tasks, such as shape reconstruction, camera pose estimation and spatiotemporal sequence reconstruction.

Strengths: Although building up correspondences with the NOCS and reasoning timespace with neural ODEs are studied by previous methods, the idea of combining them for the point cloud representation is novel. And a variety of applications indicate the promising potential of the proposed framework.

Weaknesses: The limitation section has already summarized the potential limitations of this work. I am wondering how CaSPR generalizes to objects (sequences) from unseen categories. ======================== After rebuttal: I appreciate the authors addressing my concerns.

Correctness: Yes.

Clarity: Yes. The paper is easy to follow and the illustrations are helpful to understand the main idea.

Relation to Prior Work: Yes.

Reproducibility: Yes

Additional Feedback: The idea is not surprisingly new. But the paper is well written and the extensive experiments are sufficient to demonstrate the potential of the proposed framework.

[Author Response · NeurIPS 2020]

# CaSPR: Learning Canonical Spatiotemporal Point Cloud Representations

We thank the reviewers for their comments. We appreciate that they find our work *solid* (**R2**) and *completely finished* (**R3**), our formulation *elegant* (**R1**), the CaSPR method *novel* (**All**) and *well-designed* (**R1**), the demonstrated applications *promising* (**R4**), and the paper *well-written* (**All**). In this rebuttal, we address the limitations and specific questions raised by reviewers.

**Known Category Assumption (R2, R4)**: *Training needs to be performed for the individual category which is not practical in the real world. (R2)* Indeed, for best performance CaSPR should be trained on individual categories. However, Section 2.4 of the supplement demonstrates that a single model trained on all three shape categories still gives superior results to baselines in many cases. Additionally, category-specific models can have important real world utility, e.g., in self-driving vehicles where high accuracy is critical but only for a handful of categories like cars, bicycles, and pedestrians. *How does CaSPR generalize to unseen categories? (R4)* We agree that generalization to unseen categories is interesting. Yet, this is a formidable open problem in computer vision and ML beyond the scope of our work. In CaSPR, we focus on many other problems of importance by leveraging a category-level prior on object shape.

**Novelty of Individual Components (R4)**: *Correspondences with NOCS and spacetime neural ODEs are studied previously, the idea of combining them is novel...The idea is not surprisingly new.* We emphasize that the novelty of our method is in the design and execution of its individual components – CaSPR is **not** simply a trivial combination of previous work. NOCS has been previously used to establish correspondences, but not in a temporal setting or by explicitly accounting for and normalizing time. Neural ODEs have been used to model point cloud time series in physical space (Occupancy Flow), but CaSPR does so in a learned latent space which enables more accurate temporally continuous reconstruction (see Table 4) and interesting properties like approximate latent disentanglement of shape and motion (see supplement Section 2.6). Finally, another novel contribution is TPointNet++, which facilitates operating on dynamic point cloud sequences through spatial *and* temporal canonicalization within the overall processing pipeline. Prior spatiotemporal (Occupancy Flow) and point cloud reconstruction (PointFlow) methods lack this step. We are a little confused why **R4** says the combination of these novel components is "not surprisingly new" since we are not aware of prior work that learns a spatiotemporally continuous latent representation in this way.

**Canonicalization Supervision (R1, R2)**: *The method does require supervised NOCS which can be difficult to acquire. (R1) Training requires the ground-truth for supervision, however, in practice it is difficult to obtain...is it possible to use some unsupervised criterion? (R2)* As aptly noted by **R1**, moving away from supervision in the context of NOCS was out of the scope of the current work and will be a substantial undertaking in its own right. As hinted by **R2**, there are several promising avenues that may allow the use of NOCS in real-world data. Recent work on auto-labeling NOCS in autonomous vehicle data[1] could enable direct supervised training. Alternatively, weakly-supervised approaches[2] allow learning a canonicalization with easier annotations like 2D keypoints. We will explore these in future work.

**Object vs. Scene-Level (R1, R2)**: *The method is object centric. (R1) This proposal is based on object-level sequences, it will be beneficial to generalize to scene-level. (R2)* We agree that supporting scene-level processing is a great addition and very desirable in practice; it will be a main focus of future work. Note that with recent successes in scene-level 3D segmentation and object detection, CaSPR can be applied in its current form by first localizing objects of interest.

**Move Related Work to Main Paper (R3)**: We fully agree and will integrate the extended literature overview from the supplement into the main paper for the camera-ready version.

**"Adopting" CaSPR to Applications (R3)**: *Give more details on how CaSPR is adopted to different applications.* CaSPR is a very flexible and general framework that does not require adoption to different applications: a single trained model on a category can be used for every presented application. We will make this point clearer in the paper.

**Specific Questions (R2)**: **(i)** *Why is the latent splitting reasonable? There is no mechanism that can guarantee such splitting.* We agree that CaSPR cannot guarantee a theoretical disentanglement; instead, our design encourages the canonicalization network to respect this split by only advecting the dynamic feature with the ODE. Note that this is not a CaSPR-specific drawback and many SoTA disentanglement networks rely upon the same intuition. We demonstrate experimentally that disentanglement is achieved to a large extent (see supplementary Section 2.6 for non-rigid case and video for rigid). **(ii)** *How can CaSPR guarantee it generates scene rather than deformation flow?* Since temporal correspondences naturally emerge from the CNF by using the same Gaussian noise at each timestep, there is no theoretical guarantee of scene flow. Despite this, CaSPR maintains accurate correspondences for non-rigid deformation (Table 4). If a discrepency between scene and deformation flow were to appear, a simple post-processing step such as optimal transport between generated deformations could provide temporal correspondences/scene flow. **(iii)** *Does Gaussian noise need to be the same to generate temporal correspondences?* Yes, see above.

## Footnotes

[1] *Autolabeling 3D Objects with Differentiable Rendering of SDF Shape Priors*, Zakharov *et al.*, CVPR, 2020

[2] *C3DPO: Canonical 3D Pose Networks for Non-Rigid Structure From Motion*, Novotny *et al.*, ICCV, 2019


[Meta-Review · NeurIPS 2020]

Reviewers all agreed that this is a strong submission. The proposed problem is novel and interesting. Reviewers called the method “elegant and well designed.” The paper is very well written. Reviewers also appreciated that “the experiments section does a great job at highlighting the effects of individual design choices.” The paper demonstrates the applicability of the proposed method on a number of downstream tasks. The main limitations of the paper mentioned by reviewers are: the paper builds on many methods from previous work and has limited technical novelty the method requires strong supervision that is difficult to obtain in the real world the method operates at the object level and cannot currently handle an entire scene a separate model is trained per object category Overall the reviewers all agreed that this is a very strong submission.